# Self-Erasing Network for Integral Object Attention

**Qibin Hou    Peng-Tao Jiang**
Colledge of Computer Science, Nankai University
andrewhoux@gmail.com

**Yunchao Wei**
UIUC
Urbana-Champaign, IL, USA

**Ming-Ming Cheng** *
Colledge of Computer Science, Nankai University
cmm@nankai.edu.cn

## Abstract

Recently, adversarial erasing for weakly-supervised object attention has been deeply studied due to its capability in localizing integral object regions. However, such a strategy raises one key problem that attention regions will gradually expand to non-object regions as training iterations continue, which significantly decreases the quality of the produced attention maps. To tackle such an issue as well as promote the quality of object attention, we introduce a simple yet effective **S**elf-**E**rasing **N**etwork (SeeNet) to prohibit attentions from spreading to unexpected background regions. In particular, SeeNet leverages two self-erasing strategies to encourage networks to use reliable object and background cues for learning to attention. In this way, integral object regions can be effectively highlighted without including much more background regions. To test the quality of the generated attention maps, we employ the mined object regions as heuristic cues for learning semantic segmentation models. Experiments on Pascal VOC well demonstrate the superiority of our SeeNet over other state-of-the-art methods.

## 1   Introduction

Semantic segmentation aims at assigning each pixel a label from a predefined label set given a scene. For fully-supervised semantic segmentation [21, 4, 40, 41, 20], the requirement of large-scale pixel-level annotations considerably limits its generality [3]. Some weakly-supervised works attempt to leverage relatively weak supervisions, such as scribbles [19], bounding boxes [27], or points [1], but they still need large amount of hand labors. Therefore, semantic segmentation with image-level supervision [25, 16, 35, 12, 34] is becoming a promising way to relief lots of human labors. In this paper, we are also interested in the problem of weakly-supervised semantic segmentation. As only image-level labels are available, most recent approaches [16, 3, 12, 34, 9], more or less, rely on different attention models due to their ability of covering small but discriminative semantic regions. Therefore, how to generate high-quality attention maps is essential for offering reliable initial heuristic cues for training segmentation networks. Earlier weakly-supervised semantic segmentation methods [16, 34] mostly adopt the original Class Activation Maps (CAM) model [42] for object localization. For small objects, CAM does work well but when encountering large objects of large scales it can only localize small areas of discriminative regions, which is harmful for training segmentation networks in that the undetected semantic regions will be judged to background.

Interestingly, the adversarial erasing strategy [34, 18] (Fig. 1) has been proposed recently. Benefiting from the powerful localization ability of CNNs, this type of methods is able to further discover more object-related regions by erasing the detected regions. However, a key problem of this type of

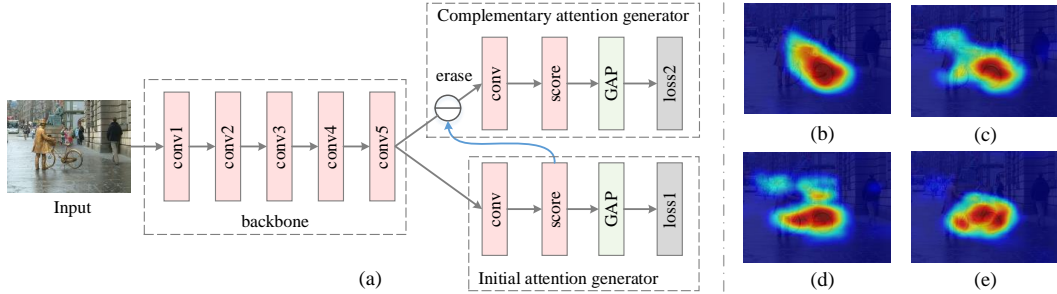

Figure 1: (a) A typical adversarial erasing approach [39], which is composed of an initial attention generator and a complementary attention generator; (b-d) Attention maps produced by (a) as the training iterations increase; (e) The attention map generated by our approach. As can be seen, the attentions by (a) gradually appear in unexpected regions while our results are confined in the bicycle region properly.

methods is that as more semantic regions are mined, the attentions may spread to the background and further the localization ability of the initial attention generator is downgraded. For example, trains often run on rails and hence as trains are erased rails may be classified as the train category, leading to negative influence on learning semantic segmentation networks.

In this paper, we propose a promising way to overcome the above mentioned drawback of the adversarial erasing strategy by introducing the concept of self-erasing. The background regions of common scenes often share some similarities, which motivates us to explicitly feed attention networks with a roughly accurate background prior to confine the observable regions in semantic fields. To do so, we present two self-erasing strategies by leveraging the background prior to purposefully suppress the spread of attentions to the background regions. Moreover, we design a new attention network that takes the above self-erasing strategies into account to discover more high-quality attentions from a potential zone instead of the whole image [39]. We apply our attention maps to weakly-supervised semantic segmentation, evaluate the segmentation results on the PASCAL VOC 2012 [6] benchmark, and show substantial improvements compared to existing methods.

## 2 Related Work

### 2.1 Attention Networks

**Earlier Work.** To date, a great number of attention networks have been developed, attempting to reveal the working mechanism of CNNs. At earlier stage, error back-propagation based methods [31, 37] were proposed for visualizing CNNs. CAM [42] adopted a global average pooling layer followed by a fully connected layer as a classifier. Later, Selvaraju proposed the Grad-CAM, which can be embedded into a variety of off-the-shelf available networks for visualizing multiple tasks, such as image captioning and image classification. Zhang *et al.* [38], motivated by humans' visual system, used the winner-take-all strategy to back-propagate discriminative signals in a top-down manner. A similar property shared by the above methods is that they only attempt to produce an attention map.

**Adversarial Erasing Strategy.** In [34], Wei *et al.* proposed the adversarial erasing strategy, which aims at discovering more unseen semantic objects. A CAM branch is used to determine an initial attention map and then a threshold is used to selectively erase the discovered regions from the images. The erased images are then sent into another CNN to further mine more discriminative regions. In [39, 18], Zhang *et al.* and Li *et al.* extended the initial adversarial erasing strategy in an end-to-end training manner.

### 2.2 Weakly-Supervised Semantic Segmentation

Due to the fact that collecting pixel-level annotations is very expensive, more and more works are recently focusing on weakly-supervised semantic segmentation. Besides some works relying on relatively strong supervisions, such as scribble [19], points [1], and bounding boxes [27], most

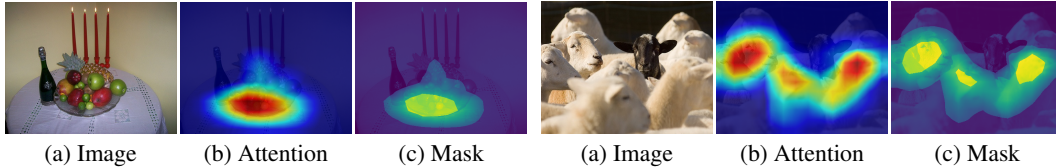

|              |               |            |              |               |            |
|:------------:|:-------------:|:----------:|:------------:|:-------------:|:----------:|
| (a) Image    | (b) Attention | (c) Mask   | (a) Image    | (b) Attention | (c) Mask   |

Figure 2: Illustrations explaining how to generate ternary masks. (a) Source images; (b) Initial attention maps produced by an initial attention generator; (c) Ternary masks after thresholding. Given (b), we separate each of them into three zones by setting two thresholds. The yellow zone in (c) corresponds to larger attention values in (b). The dark zone corresponding to lower values is explicitly defined as background priors. The middle zone contains semantic objects with high probability. Note that figures in (c) are used for explanation only but actually they are ternary masks.

weakly-supervised methods are based on only image-level labels or even inaccurate keyword [11]. Limited by keyword-level supervision, many works [16, 34, 9, 12, 3, 28, 36] harnessed attention models [42, 38] for generating the initial seeds. Saliency cues [5, 2, 33, 10, 13] are also adopted by some methods as the initial heuristic cues. Beyond that, there are also some works proposing different strategies to solve this problem, such as multiple instance learning [26] and the EM algorithm [24].

## 3 Self-Erasing Network

In this section, we describe details of the proposed **Se**lf-**E**rasing **N**etwork (SeeNet). An overview of our SeeNet can be found in Fig. 3. Before the formal description, we first introduce the intuition of our proposed approach.

### 3.1 Observations

As stated in Sec. 1, with the increase of training iterations, adversarial erasing strategy tends to mine more areas not belonging to any semantic objects at all. Thus, it is difficult to determine when the training phase should be ended. An illustration of this phenomenon has been depicted in Fig. 1. In fact, we humans always 'deliberately' suppress the areas that we are not interested in so as to better focus on our attentions [17]. When looking at a large object, we often seek the most distinctive parts of the object first and then move the eyes to the other parts. In this process, humans are always able to inadvertently and successfully neglect the distractions brought by the background. However, attention networks themselves do not possess such capability with only image-level labels given. Therefore, how to explicitly introduce background prior to attention networks is essential. Inspired by this cognitive process of humans, other than simply erasing the attention regions with higher confidence as done in existing works [39, 18, 34], we propose to explicitly tell CNNs where the background is so as to let attention networks better focus on discovering real semantic objects.

### 3.2 The Idea of Self-Erasing

To highlight the semantic regions and keep the detected attention areas from expanding to background areas, we propose the idea of self-erasing during training. Given an initial attention map (produced by $S_A$ in Fig. 3), we functionally separate the images into three zones in spatial dimension, the internal "attention zone", the external "background zone", and the middle "potential zone" (Fig. 2c). By introducing the background prior, we aim to drive attention networks into a self-erasing state so that the observable regions can be restricted to non-background areas, avoiding the continuous spread of attention areas that are already near a state of perfection. To achieve this goal, we need to solve the following two problems: (I) Given only image-level labels, how to define and obtain the background zone. (II) How to introduce the self-erasing thought into attention networks.

**Background priors.** Regarding the circumstance of weak supervision, it is quite difficult to obtain a precise background zone, so we have to seek what is less attractive than the above unreachable objective to obtain relatively accurate background priors. Given the initial attention map $M_A$, other than thresholding $M_A$ with $\delta$ for a binary mask $B_A$ as in [39], we also consider using another constant which is less than $\delta$ to get a ternary mask $T_A$. For notational convenience, we here use $\delta_h$ and $\delta_l$

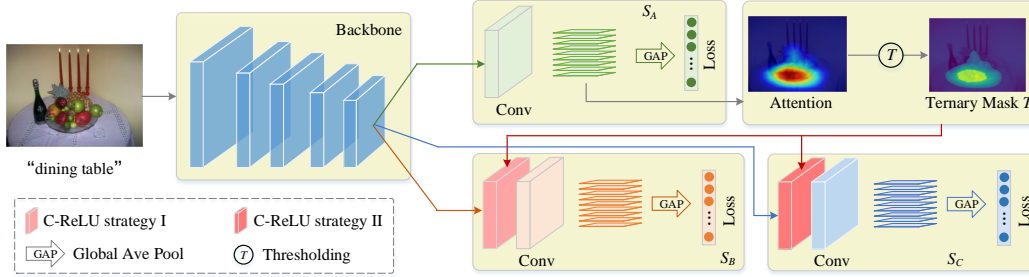

Figure 3: Overview of the proposed approach.

$(\delta_h > \delta_l)$ to denote the two thresholds. Regions with values less than $\delta_l$ in $M_A$ will all be treated as the background zone. Thus, we define our ternary mask $T_A$ as follows: $T_{A,(i,j)} = 0$ if $M_{A,(i,j)} \geq \delta_h$, $T_{A,(i,j)} = -1$ if $M_{A,(i,j)} < \delta_l$, and $T_{A,(i,j)} = 1$ otherwise. This means the background zone is associated with a value of -1 in $T_A$. We empirically found that the resulting background zone covers most of the real background areas for almost all input scenes. This is reasonable as $S_A$ is already able to locate parts of the semantic objects.

**Conditionally Reversed Linear Units (C-ReLUs).**   With the background priors, we introduce the self-erasing strategies by reversing the signs of the feature maps corresponding to the background outputted by the backbone to make the potential zone stand out. To achieve so, we extend the ReLU layer [22] to a more general case. Recall that the ReLU function, according to its definition, can be expressed as $\text{ReLU}(x) = \max(0, x)$. More generally, our C-ReLU function takes a binary mask into account and is defined as

$$\text{C-ReLU}(x) = \max(x, 0) \times B(x), \tag{1}$$

where $B$ is a binary mask, taking values from $\{-1, 1\}$. Unlike ReLUs outputting tensors with only non-negative values, our C-ReLUs conditionally flip the signs of some units according to a given mask. We expect that the attention networks can focus more on the regions with positive activations after C-ReLU and further discover more semantic objects from the potential zone because of the contrast between the potential zone and the background zone.

### 3.3   Self-Erasing Network

Our architecture is composed of three branches after a shared backbone, denoted by $S_A, S_B$, and $S_C$, respectively. Fig. 3 illustrates the overview of our proposed approach. Similarly to [39], our $S_A$ has a similar structure to [39], the goal of which is to determine an initial attention. $S_B$ and $S_C$ have similar structures to $S_A$ but differently, the C-ReLU layer is inserted before each of them.

**Self-erasing strategy I.** By adding the second branch $S_B$, we introduce the first self-erasing strategy. Given the attention map $M_A$ produced by $S_A$, we can obtain a ternary mask $T_A$ according to Sec. 3.2. When sending $T_A$ to the C-ReLU layer of $S_B$, we can easily adjust $T_A$ to a binary mask by setting non-negative values to 1. When taking the erasing strategy into account, we can extend the binary mask in C-ReLU function to a ternary case. Thus, Eqn. (1) can be rewritten as

$$\text{C-ReLU}(x) = \max(x, 0) \times T_A(x). \tag{2}$$

An visual illustration of Eqn. (2) has been depicted in Fig. 2c. The zone highlighted in yellow corresponds to attentions detected by $S_A$, which will be erased in the output of the backbone. Units with positive values in the background zone will be reversed to highlight the potential zone. During training, $S_B$ will fall in a state of self-erasing, deterring the background stuffs from being discovered and meanwhile ensuring the potential zone to be distinctive.

**Self-erasing strategy II.** This strategy aims at further avoiding attentions appearing in the background zone by introducing another branch $S_C$. Specifically, we first transform $T_A$ to a binary mask by setting regions corresponding to the background zone to 1 and the rest regions to 0. In this way, only the background zone of the output of the C-ReLU layer has non-zero activations. During the training phase, we let the probability of the background zone belonging to any semantic classes learn to be 0. Because of the background similarities among different images, this branch will help correct

**Algorithm 1:** "Proxy ground-truth" for training semantic segmentation networks

---

**Input** : Image $I$ with $N$ pixels; Image labels $\mathbf{y}$;
**Output** : Proxy ground-truth $G$
1  $Q = \text{zeros}(M+1, N)$, $N$ is the number of pixels and $M$ is the number of semantic classes;
2  $D = \text{Saliency}(I)$ ;                                          $\Leftarrow$ obtain the saliency map
3  **for** *each pixel $i \in I$* **do**
4       $A_{\mathbf{y}} = \text{SeeNet}(I, \mathbf{y})$ ;                              $\Leftarrow$ generate attention maps
5       $Q(0, i) \leftarrow 1 - D(i)$ ;                   $\Leftarrow$ probability of position $i$ being Background
6       **for** *each label $c \in \mathbf{y}$* **do**
7           $Q(c, i) \leftarrow \text{harm}(D(i), A_c(i))$ ;                   $\Leftarrow$ harmonic mean
8       **end**
9  **end**
10 $G \leftarrow \text{argmax}_{l \in \{0, \mathbf{y}\}} Q$ ;

---

the wrongly predicted attentions in the background zone and indirectly avoid the wrong spread of attentions.

The overall loss function of our approach can be written as: $\mathcal{L} = \mathcal{L}_{S_A} + \mathcal{L}_{S_B} + \mathcal{L}_{S_C}$. For all branches, we treat the multi-label classification problem as $M$ independent binary classification problems by using the cross-entropy loss, where $M$ is the number of semantic categories. Therefore, given an image $I$ and its semantic labels $\mathbf{y}$, the label vector for $S_A$ and $S_B$ is $\mathbf{l}_n = 1$ if $n \in \mathbf{y}$ and 0 otherwise, where $|\mathbf{l}| = M$. The label vector of $S_C$ is a zero vector, meaning that no semantic objects exist in the background zone.

To obtain the final attention maps, during the test phase, we discard the $S_C$ branch. Let $M_B$ be the attention map produced by $S_B$. We first normalize both $M_A$ and $M_B$ to the range $[0, 1]$ and denote the results as $\hat{M}_A$ and $\hat{M}_B$. Then, the fused attention map $M_F$ is calculated by $M_{F,i} = \max(\hat{M}_{M,i}, \hat{M}_{B,i})$. To obtain the final attention map, during the test phase, we also horizontally flip the input images and get another fused attention map $M_H$. Therefore, our final attention map $M_{final}$ can be computed by $M_{final,i} = \max(M_{F,i}, M_{H,i})$.

## 4   Weakly-Supervised Semantic Segmentation

To test the quality of our proposed attention network, we applied the generated attention maps to the recently popular weakly-supervised semantic segmentation task. To compare with existing state-of-the-art approaches, we follow a recent work [3], which leverages both saliency maps and attention maps. Instead of applying an erasing strategy to mine more salient regions, we simply use a popular salient object detection model [10] to extract the background prior by setting a hard threshold as in [18]. Specifically, given an input image $I$, we first simply normalize its saliency map obtaining $D$ taking values from $[0, 1]$. Let $\mathbf{y}$ be the image-level label set of $I$ taking values from $\{1, 2, \ldots, M\}$, where $M$ is the number of semantic classes, and $A_c$ be one of attention maps associated with label $c \in \mathbf{y}$. We can calculate our "proxy ground-truth" according to Algorithm 1. Following [3], here we harness the following harmonic mean function to compute the probability of pixel $I_i$ belonging to class $c$:

$$\text{harm}(i) = \frac{w+1}{\left(w/(A_c(i)) + 1/D(i)\right)}. \tag{3}$$

Parameter $w$ here is used to control the importance of attention maps. In our experiments, we set $w$ to 1.

## 5   Experiments

To verify the effectiveness of our proposed self-erasing strategies, we apply our attention network to the weakly-supervised semantic segmentation task as an example application. We show that by embedding our attention results into a simple approach, our semantic segmentation results outperform the existing state-of-the-arts.

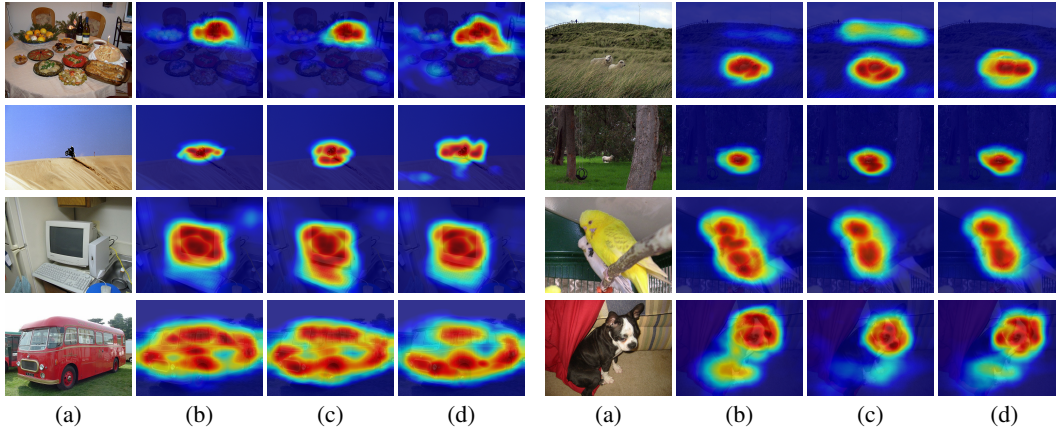

|     |     |     |     |     |     |     |     |
| (a) | (b) | (c) | (d) | (a) | (b) | (c) | (d) |

Figure 4: Visual comparisons among results by different network settings. (a) Source images; (b) Attention maps produced by our SeeNet; (c) Attention maps produced by ACoL [39]; (d) Attention maps produced by setting 2 in Sec. 5.2. The top two roles show results with small objects while the bottom two lines show results with large objects. As can be seen, our approach is able to well suppress the expansion of attentions to background regions and meanwhile generate relatively integral results compared to another two settings.

## 5.1   Implementation Details

**Datasets and evaluation metrics.** We evaluate our approach on the PASCAL VOC 2012 image segmentation benchmark [6], which contains 20 semantic classes plus the background category. As done in most previous works, we train our model for both the attention and segmentation tasks on the training set, which consists of 10,582 images, including the augmented training set provided by [7]. We compare our results with other works on both the validation and test sets, which have 1,449 and 1,456 images, respectively. Similarly to previous works, we use the mean intersection-over-union (mIoU) as our evaluation metric.

**Network settings.** For our attention network, we use VGGNet [32] as our base model as done in [39, 34]. We discard the last three fully-connected layers and connect three convolutional layers with 512 channels and kernel size 3 to the backbone as in [39]. Then, a 20 channel convolutional layer, followed by a global average pooling layer is used to predict the probability of each category as done in [39]. We set the batch size to 16, weight decay 0.0002, and learning rate 0.001, divided by 10 after 15,000 iterations. We run our network for totally 25,000 iterations. For data augmentation, we follow the strategy used in [8]. Thresholds $\delta_h$ and $\delta_l$ in $S_B$ are set to 0.7 and 0.05 times of the maximum value of the attention map inputted to C-ReLU layer, respectively. For the threshold used in $S_C$, the factor is set to $(\delta_h + \delta_l)/2$. For segmentation task, to fairly compare with other works, we adopt the standard Deeplab-LargeFOV architecture [4] as our segmentation network, which is based on the VGGNet [32] pre-trained on the ImageNet dataset [29]. Similarly to [3], we also try the ResNet version [8] Deeplab-LargeFOV architecture and report the results of both versions. The network and conditional random fields (CRFs) hyper-parameters are the same to [4].

**Inference.** For our attention network, we resize the input images to a fixed size of $224 \times 224$ and then resize the resulting attention map back to the original resolution. For segmentation task, following [20], we perform multi-scale test. For CRFs, we adopt the same code as in [4].

## 5.2   The Role of Self-Erasing

To show the importance of our self-erasing strategies, we perform several ablation experiments in this subsection. Besides showing the results of our standard SeeNet (Fig. 3), we also consider implementing another two network architectures and report the results. First, we re-implement the simple erasing network (ACoL) proposed in [39] (setting 1). The hyper-parameters are all same to the default ones in [39]. This architecture does not use our C-ReLU layer and does not have our $S_C$ branch as well. Furthermore, to stress the importance of the conditionally sign-flipping operation, we

| Settings | Training set | Supervision | mIoU (val) |
|---|---|---|---|
| 1 (ACoL [39]) | 10,582 VOC | weakly | 56.1% |
| 2 (w/o sign-flipping in C-ReLU) | 10,582 VOC | weakly | 55.8% |
| 3 (Ours) | 10,582 VOC | weakly | 57.3% |

Table 1: Quantitative comparisons with another two settings described in Sec. 5.2 on the validation set of PASCAL VOC 2012 segmentation benchmark [6]. The segmentation maps in this table are directly generated by segmentation networks without multi-scale test for fair comparisons. CRFs are not used here as well.

also try to zero the feature units associated with the background regions and keep all other settings unchanged (setting 2).

**The quality of attention maps.** In Fig. 4, we sample some images from the PASCAL VOC 2012 dataset and show the results by different experiment settings. When localizing small objects as shown on the top two rows of Fig. 4, our attention network is able to better focus on the semantic objects compared to the other two settings. This is due to the fact that our $S_C$ branch helps better recognize the background regions and hence improves the ability of our approach to keep the attentions from expanding to unexpected non-object regions. When encountering large objects as shown on the bottom two rows of Fig. 4, other than discovering where the semantic objects are, our approach is also capable of mining relatively integral objects compared to the other settings. The conditional reversion operations also protect the attention areas from spreading to the background areas. This phenomenon is specially clear in the monitor image of Fig. 4.

**Quantitative results on PASCAL VOC 2012.** Besides visual comparisons, we also consider reporting the results by applying our attention maps to the weakly-supervised semantic segmentation task. Given the attention maps, we first carry out a series of operations following the instructions described in Sec. 4, yielding the proxy ground truths of the training set. We utilize the resulting proxy ground truths as supervision to train the segmentation network. The quantitative results on the validation set are listed in Table 1. Note that the segmentation maps are all based on single-scale test and no post-processing tools are used, such as CRFs. According to Table 1, one can observe that with the same saliency maps as background priors, our approach achieves the best results. Compared to the approach proposed in [39], we have a performance gain of 1.2% in terms of mIoU score, which reflects the high quality of the attention maps produced by our approach.

## 5.3 Comparison with the State-of-the-Arts

In this subsection, we compare our proposed approach with existing weakly-supervised semantic segmentation methods that are based on image-level supervision. Detailed information for each method is shown in Table 2. We report the results of each method on both the validation and test sets.

From Table 2, we can observe that our approach greatly outperforms all other methods when the same base model, such as VGGNet [32], is used. Compared to DCSP [3], which leverages the same procedures to produce the proxy ground-truths for segmentation segmentation network, we achieves a performance gain of more than 2% on the validation set. This method uses the original CAM [42] as their attention map generator while our approach utilizes the attention maps produced by our SeeNet, which indirectly proofs the better performance of our attention network compared to CAM. To further compare our attention network with adversarial erasing methods, such as AE-PSL [34] and GAIN [18], our segmentation results are also much better than theirs. This also reflects the high quality of our attention maps.

## 5.4 Discussions

To better understand the proposed network, we show some visual results produced by our segmentation network in Fig. 6. As can be seen, our segmentation network works well because of the high-quality attention maps produced by our SeeNet. However, despite the good results, there are still a small number of failure cases, part of which has been shown on the bottom row of Fig. 6. These bad cases are often caused by the fact that semantic objects with different labels are frequently tied together,

| Methods | Publication | Supervision | mIoU (val) | | mIoU (test) |
| --- | --- | --- | --- | --- | --- |
| | | | w/o CRF | w/ CRF | w/ CRF |
| CCNN [25] | ICCV'15 | 10K weak | 33.3% | 35.3% | - |
| EM-Adapt [24] | ICCV'15 | 10K weak | - | 38.2% | 39.6% |
| MIL [26] | CVPR'15 | 700K weak | 42.0% | - | - |
| DCSM [30] | ECCV'16 | 10K weak | - | 44.1% | 45.1% |
| SEC [16] | ECCV'16 | 10K weak | 44.3% | 50.7% | 51.7% |
| AugFeed [27] | ECCV'16 | 10K weak + bbox | 50.4% | 54.3% | 55.5% |
| STC [35] | PAMI'16 | 10K weak + sal | - | 49.8% | 51.2% |
| Roy et al. [28] | CVPR'17 | 10K weak | - | 52.8% | 53.7% |
| Oh et al. [23] | CVPR'17 | 10K weak + sal | 51.2% | 55.7% | 56.7% |
| AE-PSL [34] | CVPR'17 | 10K weak + sal | - | 55.0% | 55.7% |
| Hong et al. [9] | CVPR'17 | 10K + video weak | - | 58.1% | 58.7% |
| WebS-i2 [14] | CVPR'17 | 19K weak | - | 53.4% | 55.3% |
| DCSP-VGG16 [3] | BMVC'17 | 10K weak + sal | 56.5% | 58.6% | 59.2% |
| DCSP-ResNet101 [3] | BMVC'17 | 10K weak + sal | 59.5% | 60.8% | 61.9% |
| TPL [15] | ICCV'17 | 10K weak | | 53.1% | 53.8% |
| GAIN [39] | CVPR'18 | 10K weak + sal | - | 55.3% | 56.8% |
| SeeNet (Ours, VGG16) | - | 10K weak + sal | 59.9% | 61.1% | 60.7% |
| SeeNet (Ours, ResNet101) | - | 10K weak + sal | 62.6% | 63.1% | 62.8% |

Table 2: Quantitative comparisons with the existing state-of-the-art approaches on both validation and test sets. The word 'weak' here means supervision with only image-level labels. 'bbox' and 'sal' mean that either bounding boxes or saliency maps are used. Without clear clarification, the methods listed here are based on VGGNet [32] and Deeplab-LargeFOV framework.

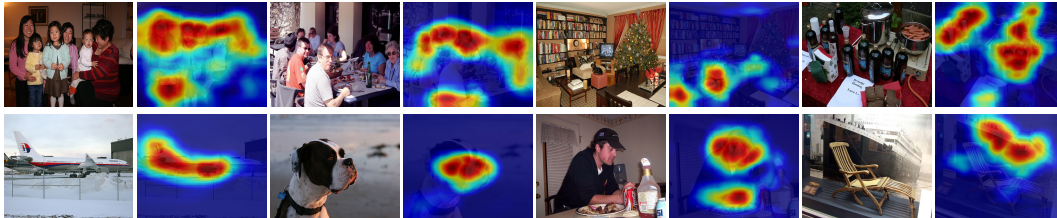

Figure 5: More visual results produced by our approach.

making the attention models difficult to precisely separate them. Specifically, as attention models are trained with only image-level labels, it is hard to capture perfectly integral objects. In Fig. 5, we show more visual results sampled from the Pascal VOC dataset. As can be seen, some scenes are with complex background or low contrast between the semantic objects and the background. Although our approach has involved background priors to help confine the attention regions, when processing these kinds of images it is hard to localize the whole objects and the quality of the initial attention maps are also essential. In addition, it is still difficult to deal with images with multiple semantic objects as shown in the first row of Fig. 5. The attention networks may easily predict which categories exist in the images but localizing all the semantic objects are not easy. A promising way to solve this problem might be incorporating a small number of pixel-level annotations for each category during the training phase to offer attention networks the information of boundaries. The pixel-level information will tell attention networks where the boundaries of semantic objects are and also accurate background regions that will help produce more integral results. This is also the future work that we will aim at.

## 6 Conclusion

In this paper, we introduce the thought of self-erasing into attention networks. We propose to extract background information based on the initial attention maps produced by the initial attention generator by thresholding the maps into three zones. Given the roughly accurate background priors, we design two self-erasing strategies, both of which aim at prohibiting the attention regions from spreading

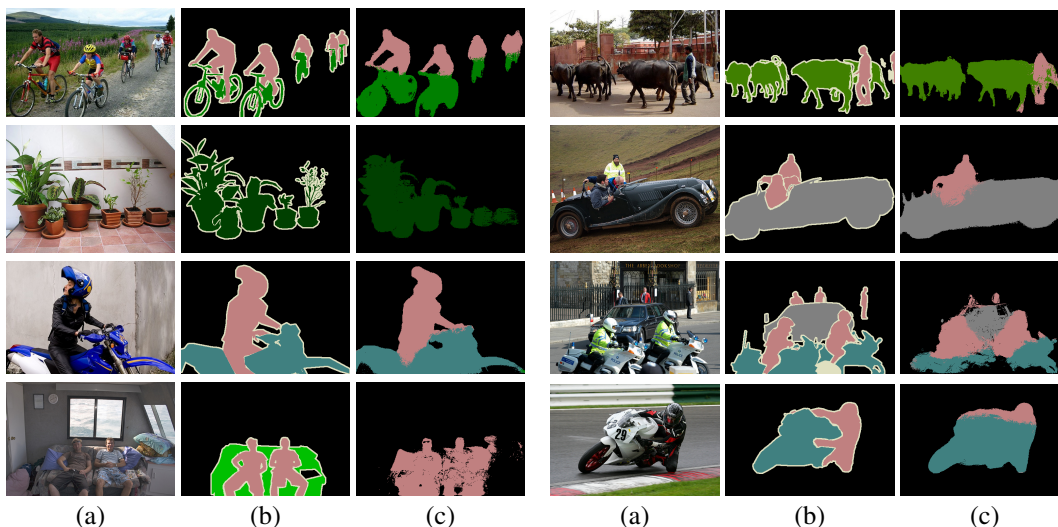

| (a) | (b) | (c) | (a) | (b) | (c) |

Figure 6: Segmentation results produced by our approach. (a) Source images. (b) Ground-truth annotations. (c) Our segmentation results. Other than good examples (the top three rows), we also list a couple of bad cases (the bottom row) to make readers better understand our work.

to unexpected regions. Based on the two self-erasing strategies, we build a self-erasing attention network to confine the observable regions in a potential zone which exists semantic objects with high probability. To evaluate the quality of the resulting attention maps, we apply them to the weakly-supervised semantic segmentation task by simply combining it with saliency maps. We show that the segmentation results based on our proxy ground-truths greatly outperform existing state-of-the-art results.

### Acknowledgments

This research was supported by NSFC (NO. 61620106008, 61572264), the national youth talent support program, Tianjin Natural Science Foundation for Distinguished Young Scholars (NO. 17JCJQJC43700), and Huawei Innovation Research Program.

## Footnotes

*MM Cheng is the corresponding author of this paper. Project page: http://mmcheng.net/SeeNet/.

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
