[Reviews · NeurIPS 2018]

Reviewer 1



Self-Handicapping Network for Integral Object Attention This paper proposes a new strategy to exploit image background information in attention networks. The use of background priors in semantic segmentation improves the estimation of attention maps. The strategy proposes to exploit a self-building ternary mask consisting of an attention, a background, and a potential zone. To maintain such mask, conditional ReLUs are proposed, which switch activations between positive and negative values depending on a mask status. This is shown to produce better contrasts between potential zones from background areas. Technically, the framework builds a 3-branched network that refines an initial attention map. Experiments are illustrated with a weakly-supervised semantic segmentation. Results indicate an improved segmentation (intersection-over-union) over recent state-of-the-art approaches, using the Pascal VOC dataset. The paper reads well, provides an exhaustive literature review on weakly supervised learning, focusing on attention network. Practical value of paper has direct high potentials in computer vision applications. Experiments demonstrate superiority of approach with respect to recent state-of-the-art. No obvious flaws in manuscript. Possibly on the negative side, performance is higher but arguably on the same range of competitive approaches, perhaps showing a significance test on the segmentation results could give a stronger argument. If correct, "self-handicapping" means here "self-erasing". This latter terminology may perhaps be better related with the current literature on adversarial erasing methods? (I am aware of "erasing" networks, but not on any "handicapped" networks) The authors' feedback proposes to add a figure with best and worst visual results - taking half a page - is a 9th page allowed? if not, the rebuttal loose its value - Furthermore, as mentioned in the review, results are in the same range as competing methods - authors missed addressing this question, on whether the proposed results consistently outperform other method (statistical significance of their results) - they only provides average accuracies - unless this was misunderstood from my side?

Reviewer 2



This paper proposes a self-handicapping network based on a new adversarial erasing strategy for object attention. The motivation of the proposal is clear and the method is well introduced. The contribution is clear and sufficient. The experimental results properly validates the improvement achieved with the proposed technique in the image segmentation task. Moreover, a comparison with weakly-supervised semantic segmentation methods of the state of the art is included. In my opinion, the paper can be accepted as it is.

Reviewer 3



Self-Handicapping Network for Integral Object Attention The paper studies the problem of generating an object-level attention map using image-level supervision. This is a fundamental problem and is helpful for understanding deep CNNs. However, the problem is very challenging since only weak supervision is given. The paper follows the work of adversarial erasing [35] for weakly-supervised object attention, which iteratively mines object pixels by erasing the detected object regions. However, when stopping the erasing process is a problem. Thus, the paper proposes a self-handicapping strategy. The main idea is to fully utilize background prior. It designs a new attention network in which the background prior can revise the sign of feature map to constrain the discovering of object attention in a zone of potential instead of the whole image. Pros: + It propose a new self-handicapping network which extents the original CAM network with multiple branches. By threshing the score map generated by CAM with two thresholds, it obtains a ternary map acts as a switch to edit the feature maps in the following branches. The ternary map can both help to suppress background prior and erasing the mined foreground. It is an elegant design. + The method has been proven to be effective in solving the weakly-supervised semantic segmentation problem. Testing in the Pascal VOC dataset, the proposed ShaNet outperforms some recent methods. In addition, the effectiveness of self-handicapping has been visually illustrated. + The paper is well-written and easy to follow. Cons: - In figure 3, I can not observe the difference between the branches of S_B and S_C. However, from the description in Self-handicapping strategy I and II, the C-ReLU functions are different in the two branches. Thus, figure 3 should be improved. - The reason why S_C is ignored in testing is not given. The authors should justify why S_C is helpful in training while not helpful in testing.